# Optimal Intervention Timing for Robotic-Assisted Gait Training in Hemiplegic Stroke

**DOI:** 10.3390/brainsci12081058

**Published:** 2022-08-10

**Authors:** Lingchao Xie, Bu Hyun Yoon, Chanhee Park, Joshua (Sung) H. You

**Affiliations:** 1Sports Movement Artificial Robotics Technology (SMART) Institute, Department of Physical Therapy, Yonsei University, Wonju 26493, Korea; 2Department of Physical Therapy, Yonsei University, Wonju 26493, Korea

**Keywords:** stroke, recovery stage, robotic-assisted gait training, Walkbot, sensorimotor function

## Abstract

This study was designed to determine the best intervention time (acute, subacute, and chronic stages) for Walkbot robot-assisted gait training (RAGT) rehabilitation to improve clinical outcomes, including sensorimotor function, balance, cognition, and activities of daily living, in hemiparetic stroke patients. Thirty-six stroke survivors (acute stage group (ASG), *n* = 11; subacute stage group (SSG), *n* = 15; chronic stage group (CSG), *n* = 10) consistently received Walkbot RAGT for 30 min/session, thrice a week, for 4 weeks. Six clinical outcome variables, including the Fugl–Meyer Assessment (FMA), Berg Balance Scale (BBS), Trunk Impairment Scale (TIS), Modified Barthel Index (MBI), Modified Ashworth Scale (MAS), and Mini-Mental State Examination, were examined before and after the intervention. Significant differences in the FMA, BBS, TIS, and MBI were observed between the ASG and the SSG or CSG. A significant time effect was observed for all variables, except for the MAS, in the ASG and SSG, whereas significant time effects were noted for the FMA, BBS, and TIS in the CSG. Overall, Walkbot RAGT was more favorable for acute stroke patients than for those with subacute or chronic stroke. This provides the first clinical evidence for the optimal intervention timing for RAGT in stroke.

## 1. Introduction

An extensive review of current pieces of evidence for treadmill-based stationary exoskeletal robotic-assisted gait training (RAGT) suggests that this intervention has beneficial effects on sensorimotor function, balance, gait, and performance of activities of daily living (ADLs) in acute [1,2,3], subacute [4,5,6], and chronic [7,8,9] stroke patients. Currently, the Walkbot ankle–knee–hip-controlled RAGT model (P&S Mechanics, Seoul, Republic of Korea) is commonly used, and its clinical efficacy has been well investigated in the acute, subacute, and chronic stages of recovery [2,6,10]. However, until now, only one clinical trial has reported that Walkbot-based interventions have more positive effects on balance (20.4%) on the Berg Balance Scale (BBS), and ambulation ability (23.00%) in the Functional Ambulation Category (FAC), than conventional physical therapy (CPT) in the acute stage of stroke rehabilitation [2]. Moreover, previous clinical studies have reported inconsistent results [1,10,11,12] for RAGT in the acute, subacute, and chronic stages of stroke rehabilitation. Chang et al. observed that exoskeleton-device-based RAGT has positive effects on sensorimotor function (16.18%) in 20 acute hemiparetic stroke patients [1], whereas Taveggia et al. showed no significant effects on gait in the 6-min walk test in 13 subacute stroke patients [11]. Michiel et al. reported that exoskeleton-device-based RAGT has more positive effects on balance (38.75%) and ambulation ability (52.00%) in the FAC than CPT alone in the chronic stage [12]. However, whether such treadmill-based stationary exoskeletal RAGT has any differences in the recovery of the sensorimotor function, balance, gait, and ADL performance between the acute [1,2,3], subacute [4,5,6], and chronic [7,8,9] stages in stroke patients are unknown. Hence, the clinical decisions upon which stage of recovery RAGT should be prescribed and implemented to obtain optimal results remain controversial.

The Cochrane review combined all previous RAGT studies, and found that in 24 randomized controlled trials (RCTs) involving 1243 participants, the probability of walking independently in the acute or subacute phase of stroke using RAGT increased. In contrast, in 16 RCTs involving 461 participants, the chance of walking independently in the chronic phase of stroke did not increase [13]. In a proof-of-concept study, throughout a critical period of heightened neuroplasticity, rats that suffered stroke were subjected to an enriched environment combined with daily sessions of grasping training. The most substantial improvement in forelimb reaching ability recovery was achieved when enriched rehabilitation (6 h/session) was initiated early, 5 days after stroke, compared to 14 and 30 days after stroke. Furthermore, recovery was correlated with increased dendritic branching of layer V motor cortex neurons in the intact hemisphere, which is a response that was not detected when rehabilitation was delayed by 30 days [14]. This experimental evidence plausibly suggests that individuals in the acute and subacute phases of stroke may benefit from RAGT more than those in the chronic phase. However, because of the limitations and uncertainty of the evidence, further investigation into determining the optimal intervention timing for initiating RAGT after stroke, and confirming how the effects are affected by the passage of time, is highly desirable.

Certainly, there is a clear need to determine the optimal and effective stage for RAGT treatment at which the best effects on sensorimotor function, balance, trunk stability, and ADL performance are achieved during the acute, subacute, and chronic stages in hemiplegic stroke patients. Therefore, this study was designed to compare the effects of Walkbot RAGT on sensorimotor recovery in the Fugl–Meyer Assessment (FMA), balance function in the BBS, trunk stability in the Trunk Impairment Scale (TIS), ADL performance in the Modified Barthel index (MBI), spasticity in the Modified Ashworth Scale (MAS), and cognitive function using the Mini-Mental State Examination (MMSE) between the acute, subacute, and chronic phases of stroke recovery. We hypothesized that patients in the acute stage would achieve more benefits to sensory motor function, balance, ADL performance, and trunk stability in the short term using exoskeletal Walkbot RAGT. However, this recovery rate would slow down over time, reaching a slow but smooth level in the chronic phase. Although the ability of RAGT to enhance cognitive function may be limited, we hypothesized that it could influence cognitive function.

## 2. Materials and Methods

### 2.1. Participants

The electronic rehabilitation dataset system of the hospital was used to access the medical records of the patients. The inclusion criteria were as follows: (1) stroke survivors; (2) ambulators who depend on assistance (FAC between 2 and 4); (3) those aged 18–99 years; (4) those with a height of 132–200 cm; (5) those who were eligible for gait training (i.e., those who can walk at least one step with apparatus/assistance, as clinically assessed); (6) those with a hip–knee length of 33–48 cm; and (7) those with a knee–foot length of 33–48 cm. The exclusion criteria were as follows: (1) patients weighing > 135 kg; (2) those with uncontrolled stage 2 hypertension with blood pressure > 160/100 mmHg; (3) those with cardiopulmonary dysfunction that affects the ambulation test; (4) those with periosteal dysfunction, such as skin breakdown or decubitus ulcers around the weight-bearing area of the suspensory belt; (5) those with significant and persistent mental illness; (6) those with lower extremity fixed contracture or deformity; (7) those with bone instability, comprising non-combined fractures, unstable spine, or severe osteoporosis; (8) those with other neurodegenerative disorders, including Parkinson’s disease and amyotrophic lateral sclerosis; (9) those with MAS > 3 of the affected limb; (10) those with significant pain and sensory deficits; and (11) those with aphasia and dysarthria that affect their ability to communicate discomfort [2]. The group classification of acute (<1 week), subacute (2–24 weeks) and chronic (>24 weeks) stages after stroke was performed based on Grefkes’s post-stroke onset time reference [15].

### 2.2. Study Design

This retrospective study was designed to determine the clinical outcomes before and after the Walkbot RAGT intervention in the acute stage group (ASG) (<1 week), subacute stage group (SSG) (2–24 weeks), and chronic stage group (CSG) (>24 weeks) [15]. The grouping assignment was based on the stroke recovery stage. Standardized clinical outcome tests, including the FMA, BBS, TIS, MBI, MAS, and MMSE, were performed before and after the intervention. The retrospective study was approved by the Institutional Review Board and Ethics Committee of Chungdam Hospital (IRB number: CDIRB-2021-004), and registered at the International Clinical Trials Registry Platform (KCT0006333). Demographic and clinical information was collected from all patients. Figure 1 shows the study flowchart.

#### 2.2.1. Sensorimotor Function Assessment

The FMA is designed to assess motor function, balance, sensation, and joint function in hemiplegic stroke patients. The FMA is used clinically and non-clinically to determine disease severity. This tool contains items assessed using a three-point ordinal scale. If the patients could not complete the task, the item is scored 0. A score of 1 is given for partial completion, and a score of 2 is given for full completion. The reliability and validity of the outcome measures in stroke patients were *r* = 0.98 and *r* = 0.96, respectively [16].

#### 2.2.2. Balance Measurement

The BBS is used to objectively determine a patient’s ability to maintain balance during a series of predetermined tasks. It is a 14-item list, with each item assessed using a five-point ordinal scale, ranging from 0 (unable to perform the task) to 4 (able to perform the task independently). All scores were added to the final score. The reliability and validity of the outcome measures were *r* = 0.98 and *r* = 0.90, respectively [17].

#### 2.2.3. Trunk Coordination

The TIS is used to measure trunk motor impairment after stroke by evaluating static and dynamic sitting balance and trunk movement coordination. Scores range from 0 to 23. A two-, three-, or four-point ordinal scale is used. The reliability and validity of the outcome measurement tests were *r* = 0.91 and *r* = 0.83, respectively [18].

#### 2.2.4. ADLs

The MBI is an ordinal scale used to evaluate a patient’s ability to perform ADLs. The MBI consists of 10 variables that depict self-care and mobility, with higher scores indicating a greater ability to function independently after discharge. The time and physical assistance required to perform each item were used to determine the assigned value for each item. The reliability and validity of the outcome measurement tests were *r* = 0.94 and *r* > 0.92, respectively [19].

#### 2.2.5. Muscle Tone Assessment

The MAS is used to measure resistance during passive soft tissue stretching and is used as a simple measure of spasticity in hemiparetic stroke patients. It is performed by first extending the patient’s limb from the maximal flexion possible to the maximal extension possible. Subsequently, the MAS is assessed as the subject moves from extension to flexion. The lowest score is 0 (no muscle tone), and the highest score is 4 (limb rigidity during flexion or extension). The reliability and validity of the outcome measures were *r* = 0.84 and *r* = −0.94, respectively [20]. In this study, knee flexor muscle spasticity was assessed.

#### 2.2.6. Cognitive Function Assessment

The MMSE is used to measure cognitive impairment and function, estimate the severity and progression of cognitive impairment, and follow up cognitive dementia changes over time. It includes a 30-point questionnaire, including questions on orientation, attention, memory, language, and visual–spatial skills. The validity and reliability of the MMSE were *r* = 0.88 and *r* = 0.86, respectively [21].

### 2.3. Intervention

The three groups received therapy for 30 min/session, thrice a week, for 4 weeks (12 sessions in total) [22], without any preparation time (Figure 1). As illustrated in Figure 2, the Walkbot system has a built-in hip–knee–ankle actuator that offers an optimal hip–knee–ankle joint motion trajectory during walking [23]. The adjustable leg length and ankle range of motion control allow the Walkbot system to create movements as close as possible to human kinematics and dynamics [24]. This robotic system is designed to detect a patient’s ongoing gait characteristics, that is, the amount of participation or use of active joint angular displacement excursion, active weight-bearing center of pressure, and active force/torque. This system also provides real-time feedback, allowing precision in ankle–knee–hip kinematics and kinetics. Specifically, Walkbot RAGT can provide proper kinematic, kinetic, and proprioceptive guidance; variable error practice; and high-intensity, interactive, repetitive, and task-specific paretic lower limb exercises [25].

The thigh and calf lengths were measured before the harness was attached. An initial counterweight (30%), which was gradually reduced according to the patient’s length, was used. The RAGT body weight support (BWS) was set to 100% and progressively decreased until the knee began to flex during the standing phase. Throughout the sessions, the physiotherapist controlled the BWS and monitored the knee condition. The gait speed was initially set at 1.0 km/h and adjusted to the patient’s comfortable pace, and progressed to 1.8 km/h [2,26]. Experienced physical therapists provided the standardized verbal encouragements and feedback about gait performance as needed during the RAGT intervention for all patients [27].

### 2.4. Statistical Analyses

The results are expressed as means and standard deviations. Power analysis using G*Power (version 3.1.9.4; Franz Faul, University of Kiel, Kiel, Germany) was performed to determine the minimum sample size required. Based on our previous study, the sample size was determined to be 30–36 according to an effect size of eta squared (*η*^2^) of 0.6 and a power 1-*β* of 0.8 using the FMA, BBS, TIS, MBI, and MAS [2,28,29]. All continuous variables for the three groups were analyzed separately using the Kolmogorov–Smirnov test to test the assumption of a normal distribution. For the demographic data from the three groups, analysis of variance (ANOVA) was used for continuous variables. Two-way repeated ANOVA was used to compare the pre–post difference between the acute, subacute, and chronic groups, and Bonferroni’s post hoc test was used to account for type I errors. Analysis of covariance (ANCOVA) was used for intergroup comparisons because the baseline (pre-test) FMA outcome measurement was statistically different among the groups. A pre–post comparison was performed using a paired *t*-test. Statistical Package for the Social Sciences (version 25.0; IBM Corp., Armonk, NY, USA) was used to perform statistical analyses. Statistical significance was set at *p* < 0.05.

## 3. Results

A convenience sample of 36 hemiparetic stroke patients (mean age: 66.86 ± 11.51 years; 16 females) who were admitted and received the standardized Walkbot RAGT retrospective study protocol between July 2018 and July 2020 at the rehabilitation hospital were retrospectively evaluated. Because of the complicated nature of stroke patients, only 36 (76%) of the 47 patients who successfully completed the pre-test, intervention, and post-test were included in the final data analysis. In total, 11 patients were excluded due to missing data or incomplete data associated with sudden discharge and medical complications. Patient demographics and clinical characteristics are shown in Table 1. The independent *t*-test did not reveal any significant differences in the baseline demographic and clinical gait impairment characteristics, including FAC, among the three groups, indicating group homogeneity.

### 3.1. Sensorimotor Function

ANCOVA showed significant improvements in the FMA (*p* = 0.02) between the acute, subacute, and chronic stages (Table 2). Bonferroni’s post hoc test confirmed that the ASG showed a greater increase in the FMA than the SSG (*p* = 0.025) (Figure 3), although other intergroup comparisons were not significantly different.

### 3.2. Balance Measurement

ANOVA showed significant changes in the BBS (*p* = 0.03) between the acute, subacute, and chronic stages (Table 2). Bonferroni’s post hoc analysis revealed that the ASG showed a greater increase in the BBS than the SSG (*p* = 0.028) (Figure 3), although other intergroup comparisons were not significantly different.

### 3.3. Trunk Coordination

ANOVA showed significant differences in the TIS (*p* = 0.03) between the ASG, SSG, and CSG (Table 2). Additionally, Bonferroni’s post hoc test revealed that the ASG showed a greater increase in the TIS than the CSG (*p* = 0.029), although other intergroup comparisons were not significantly different (Figure 3).

### 3.4. ADLs

ANOVA showed significant changes in the MBI (*p* = 0.02) between the acute, subacute, and chronic stages (Table 2). Furthermore, Bonferroni’s post hoc test revealed that the ASG showed a greater increase in the MBI than the CSG (*p* = 0.018), although other intergroup comparisons were not significantly different (Figure 3).

### 3.5. Muscle Tone Assessment

The paired *t*-test did not show any pre–post changes, while ANOVA did not show any significant differences in the MAS score (*p* = 0.43) between the ASG, SSG, and CSG (Table 2), suggesting that spasticity did not significantly change after the Walkbot RAGT intervention.

### 3.6. Cognitive Function

ANOVA did not show significant changes in cognitive function (Table 2), as assessed using the MMSE, between the ASG, SSG, and CSG (*p* = 0.88) (Figure 3). Moreover, Bonferroni’s post hoc test failed to reveal any significant differences in the mean MMSE scores between the three groups.

## 4. Discussion

To the best of our knowledge, this study is the first to compare the effects of the Walkbot RAGT intervention on sensorimotor recovery, balance function, trunk stability, ADL performance, spasticity, and cognitive function between the acute, subacute, and chronic stages of stroke. Consistent with our hypothesis, the exoskeletal Walkbot RAGT produced more notable improvements in the recovery of sensorimotor function, balance, ADL performance, and trunk coordination in the acute stage than in the other two stages. Most importantly, the FMA, BBS, TIS, and MBI clinical outcomes were statistically different between the ASG and the other two groups. Simultaneously, the pre–post comparison showed that patients in the ASG and SSG demonstrated statistically significant improvements in all variables, except for the MAS, after the Walkbot RAGT intervention. In contrast, patients in the CSG only showed statistically significant improvements in three outcome measures after the RAGT intervention, namely, the FMA, BBS, and TIS. Nevertheless, the lack of similar studies makes it difficult to compare our findings with those of previous studies related to the optimal and effective intervention time for RAGT.

Sensorimotor function analysis revealed significant improvements in the FMA scores (mean difference, 4.31%) in all three groups, and the post hoc analysis showed statistically significant differences between the ASG and SSG (*p* < 0.05), which suggests greater improvements in sensorimotor recovery after the Walkbot RAGT intervention in the ASG. This is consistent with the results of previous studies on exoskeleton RAGT in improving sensorimotor function in hemiparetic stroke [9,28,30]. In 2012, Chang et al. reported significant improvements in the FMA score (16.18%) in 20 acute hemiparetic stroke patients [1]. In addition, Oh et al. (2021) reported that the FMA (3.78%) improved after 6 weeks of RAGT in 57 subacute hemiparetic stroke patients [28], whereas Kim et al. (2020) found that the recovery of sensorimotor function improved after 4 weeks of RAGT (2.86%) in 14 chronic hemiparetic stroke patients [9]. Rhythmic, repetitive, and concentrated gait training is effective in improving lower limb motor function in stroke patients. Studies have suggested that approximately 300–500 repetitions are required to improve the recovery of lower limb motor function [31,32]. Mackay et al. [33] investigated the effect of moderate-to-vigorous aerobic exercise on brain-derived neurotrophic factor, growth hormone, and cortisol levels, and/or changes in neurotransmitters, as well as the immediate and rapid error reduction capability in a precision moving cursor task, in 20 chronic stroke patients, and reported beneficial exercise-induced neuroplastic and motor behavioral results. Hence, these neurophysiological and motor performance improvements support the key role of intensive movement utilization or paretic limb exercise in forming new neurons, developing and strengthening existing or spared neurons, and cortically reorganizing neuronal substrates [34,35,36]. Our recent electro-encephalography (EEG) study found that a cortical neuroplasticity change was associated with the improvement in the recovery of motor function and FMA scores of hemiparetic stroke patients, further corroborating locomotor recovery as a function of RAGT [30].

An analysis of a clinical balance test showed significant improvements in the BBS (mean difference, 9.78%) after RAGT, and the post hoc analysis revealed a significant difference between the acute and subacute stages (*p* < 0.05), which suggests that the improvement in the BBS in the ASG was superior to that in the SSG and CSG. These results are consistent with those reported in previous Walkbot RAGT experimental studies, indicating that this intervention has beneficial effects on balance [2,37]. For example, Park et al. (2020) revealed that Walkbot RAGT significantly increased the BBS scores (20.4%) after 3 weeks in 58 acute hemiparetic stroke patients [2]; Kim et al. (2016) showed BBS score improvements (14.09%) after RAGT intervention in 38 subacute stroke patients [5]; and Bang et al. (2016) reported BBS score improvements (10.13%) in 14 chronic stroke patients after RAGT intervention [7]. A plausible potential mechanism for this beneficial enhancement is that the Walkbot RAGT system provides weight support, repetitive motion, tactile guidance, and proprioceptive and somatosensory feedback, resulting in promising improvements in balance [38]. The most significant improvement in the ASG (Figure 3) was consistent with our hypothesis, demonstrating that RAGT in the acute stage can dramatically improve the balance ability of patients, and provide a foundation for subsequent recovery.

Furthermore, we found significant changes in the TIS (mean difference, 7.91%) after RAGT intervention in all groups; specifically, the difference between the ASG and CSG was statistically significant (*p* < 0.05), which indicates greater improvements in trunk coordination in the ASG. This finding is consistent with that reported in our previous RAGT study involving subacute stroke patients, which revealed remarkable TIS (16.74%) enhancement [28]. Yoon et al. (2022) also demonstrated TIS improvement (17.78%) in 32 hemiparetic stroke patients after RAGT intervention [39]. Likewise, Kim et al. (2019) reported the positive effect of RAGT on the TIS (6.74%) in 19 chronic stroke patients [40]. Hemiplegic stroke patients have reduced joint mobility and inadequate forward propulsion, leading to asymmetrical and unstable trunk stability during standing and walking tasks. Such asymmetrical and unstable trunk stability during standing, as well as walking performance, improved after Walkbot RAGT intervention [41]. Figure 3 illustrates that such improvement tends to diminish across the acute, subacute, and chronic stages, demonstrating that the motor recovery effect provided by RAGT decreases over time post-stroke. The same exercise dosage has more effective benefits in the early phase of rehabilitation.

ADL analysis showed significant improvements in the MBI (mean difference, 7.27%) in the ASG and SSG after RAGT intervention. The post hoc analysis indicated a statistically significant difference in the MBI between the ASG and CSG (*p* < 0.05), which demonstrates more improvement in ADL performance in the ASG. This is consistent with the results of an earlier RAGT study involving hemiparetic stroke patients, which revealed a greater improvement in the MBI (21.92%) in the acute group than in the remaining groups [25]. Chung et al. (2017) also demonstrated increased ADL performance (18.9%) in 14 acute stroke patients after RAGT intervention [37]. Schwartz et al. (2009) indicated that RAGT increased ADL performance (10.81%) in 67 subacute hemiparetic stroke patients [6]. Cho et al. (2015) reported that RAGT has positive effects on the MBI (8.70%) in 20 chronic stroke patients [8]. More functional categories showed significant improvements in patients in the acute group than in those in the other groups. Therefore, in the same 4 weeks, the recovery of all functions was faster, and the improvement in ADL performance was naturally greater, in the acute stage than in the other two stages.

Cognitive function analysis revealed some improvements in the MMSE scores in the ASG (3.94%) and SSG (4.05%); however, this improvement did not appear to be related to the intervention timing (*p* > 0.05). This is consistent with the results of a previous RAGT study, which showed improvements in the MMSE scores (16.67%) in chronic hemiparetic stroke patients [42]. Dundar et al. (2014) also reported an increase in the MMSE scores (8.00%) in 36 chronic stroke patients after RAGT intervention [43]. A possible underlying neurophysiological mechanism is that cognitive changes, including attention, memory, and processing speed, in the anterior cingulate cortex (ACC) and hippocampal regions may be correlated, as these regions represent tightly connected neighborhood network nodes that may be positively influenced by intensive RAGT, walking exercise, or task-specific movement [44], supporting the idea that the connection between the ACC and hippocampus improved after exercise [45]. These findings, combined with the results of this study, may help maximize the recovery of patients in the acute stage of stroke recovery as a function of RAGT, thus enhancing their subsequent recuperation and preventing cognitive deterioration.

Current RAGT research primarily focuses on the effectiveness of Walkbot RAGT in improving patients’ motor function compared with the effectiveness of CPT. However, to maximize post-stroke rehabilitation and clinical cost-effectiveness, determining the exact time and stage at which the stroke recovery response for robotic neurorehabilitation is crucial [46]. This finding further corroborates our results that acute stroke patients demonstrate more meaningful improvements when receiving the same RAGT interventional dosages as subacute or chronic stroke patients [30,47]. In a Cochrane review, Mehrholtz et al. suggested that the superiority of locomotor recovery tends to be greater during the subacute stage than during the acute stage, which may involve the spontaneous recovery mechanism following RAGT [13]. In contrast, Chollet et al. investigated motor recovery in 113 acute ischemic stroke patients, who were randomly assigned either to a group that received oral fluoxetine (20 mg/day) or a group that received a placebo pill, for 3 months, and reported that the oral fluoxetine group had significantly greater improvement in the FMA than the placebo group [48]. In this study, the underlying therapeutic effects of fluoxetine and neurotransmitters enhanced motor recovery outcomes. While the therapeutic application in humans further explains the neurophysiological underpinnings of post-stroke treatment-driven and spontaneous locomotor recovery mechanisms, deciphering the precise neurophysiological recovery mechanisms similar to those in animals is extremely difficult and challenging, even with advanced neuroimaging techniques, such as functional magnetic resonance imaging, positron emission tomography, single-photon emission computed tomography, EEG, magnetoencephalography, transcranial magnetic stimulation, and near-infrared spectroscopy [49]. Nevertheless, we have previously investigated the effects of the combination of CPT and RAGT on synergy (10.6%) and spasticity (4.00%) and compared them with those of CPT alone in acute stroke patients [49]. We found that the spontaneous recovery mechanism was further augmented by the therapy-induced recovery mechanism that results from the combination of CPT and RAGT in acute stroke patients [29]. Moreover, our previous RAGT randomized controlled trial demonstrated that the combination of CPT and RAGT had more beneficial effects on the recovery rate difference of ambulation (10.73%), cardiopulmonary function (5.00%), balance (28.92%), and fall confidence (87.61%) than CPT alone among acute hemiparetic stroke patients, possibly supporting the assumption of augmented overlapping of the treatment-driven and spontaneous mechanisms [2].

The present study has a couple of research limitations that should be considered in the future. One limitation is that this study used a non-randomized design because the grouping assignment had to be made according to the recovery stage of the subject population. Second, tools for measuring gait dependency/disability, such as the FAC possesses, were not used. Another limitation is that the long-term recovery of acute and subacute stroke patients was not ascertained to determine whether any further improvements could be obtained. In addition, since both ischemic and hemorrhagic strokes were included in all groups of this study, and ischemic stroke is generally considered to have a better prognosis than hemorrhagic stroke, the results should be interpreted cautiously. Nevertheless, our baseline ANOVA data showed no statistical difference in the type of stroke variable across the groups, indicating that the stroke type is less likely to be confounded. This study recommends conducting further studies to determine whether such improvements over an extended period can be achieved by conducting a corresponding comparative analysis to yield more meaningful findings regarding the optimal timing for effective and sustainable RAGT intervention.

## 5. Conclusions

This study demonstrated that RAGT improves the recovery of sensorimotor function, balance, ADL performance, and trunk stability more effectively in the acute stage than in the subacute and chronic stages of stroke. Our findings provide clinical evidence-based insights for determining the most appropriate time at which rehabilitation interventions should be performed to achieve the maximal recovery of sensorimotor function, balance, ADL performance, and trunk stability.

## Figures and Tables

**Figure 1 brainsci-12-01058-f001:**
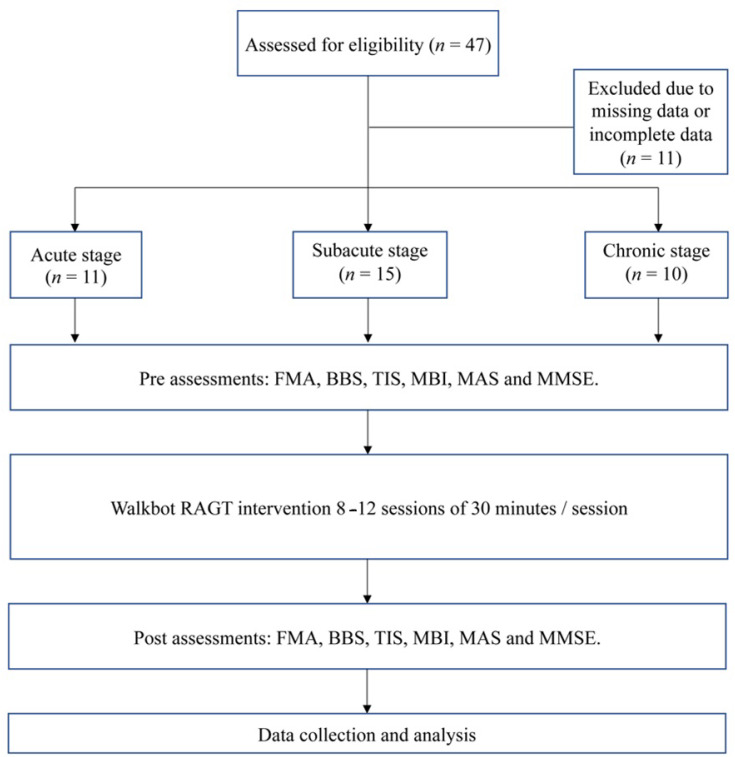
Flowchart of the study.

**Figure 2 brainsci-12-01058-f002:**
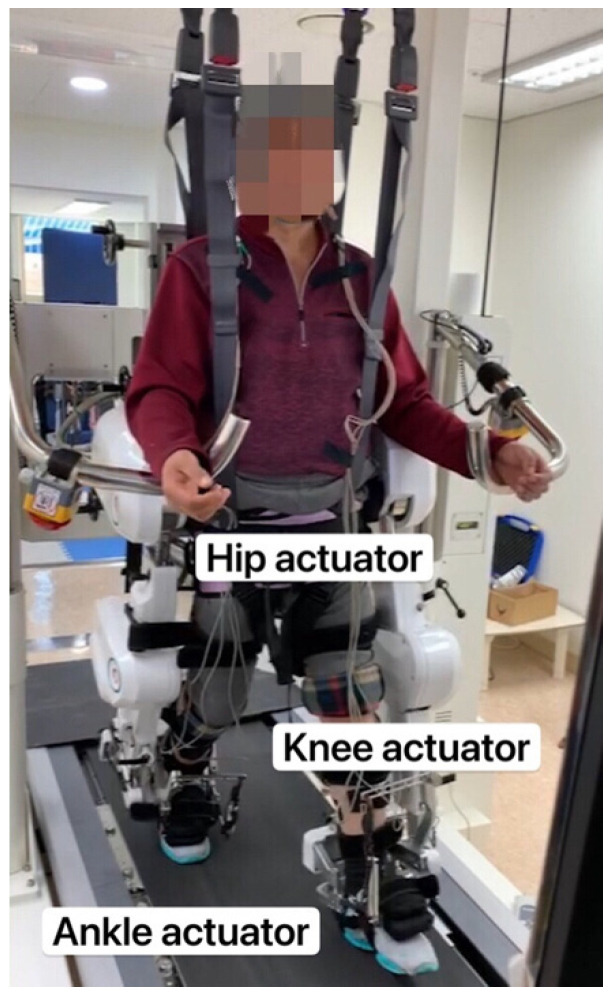
Walkbot system.

**Figure 3 brainsci-12-01058-f003:**
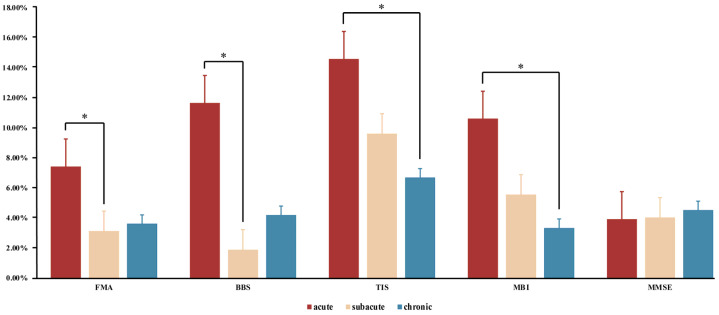
Clinical outcome variables after Walkbot RAGT in acute, subacute, and chronic stroke patients. * *p* < 0.05, Bonferroni’s post hoc test.

**Table 1 brainsci-12-01058-t001:** Baseline demographic and clinical characteristics of the patients (*N* = 36).

Characteristics	Acute Stage Group (*n* = 11)	Subacute Stage Group (*n* = 15)	Chronic Stage Group (*n* = 10)	*p*-Value *
Age (years) ^a^	63.18 ± 12.61	68.33 ± 11.29	68.43 ± 9.43	0.55
Sex (male/female)	5/5	10/4	3/7	0.45
Height (cm) ^a^	164.18 ± 10.63	162.13 ± 7.46	168.00 ± 9.75	0.49
Weight (kg) ^a^	65.14 ± 8.63	61.29 ± 7.70	65.50 ± 13.61	0.21
Hemiparetic side (left/right)	3/7	8/7	7/3	0.52
FAC ^a^	2.82 ± 0.57	3.07 ± 0.57	3.10 ± 0.54	0.38
FMA ^a^ pre-baseline	30.55 ± 20.93	13.13 ± 10.19	19.40 ± 2.01	0.01 *
Type of stroke (Ischemic/Hemorrhagic)	7/3	6/9	4/6	0.71

FAC, Functional Ambulation Categories; FMA, Fugl–Meyer assessment. ^a^ Mean ± standard deviation. * One-way ANOVA at *p* < 0.05.

**Table 2 brainsci-12-01058-t002:** Clinical outcome measure differences between before and after Walkbot RAGT interventions in the acute, subacute, and chronic stages of stroke (*N* = 36).

		FMA	BBS	TIS	MBI	MAS	MMSE
Acute stage group	Pre-test	30.55 ± 20.93	7.45 ± 5.58	5.27 ± 5.63	36.01 ± 13.34	0.27 ± 0.44	24.73 ± 4.22
	Post-test	38.00 ± 21.99	14.00 ± 11.65	8.64 ± 6.08	47.55 ± 15.10	0.27 ± 0.62	25.91 ± 3.53
	Mean change	7.45 ± 5.58	6.55 ± 7.98	3.36 ± 1.67	10.64 ± 7.88	0.00 ± 0.43	1.18 ± 1.64
	*p*-value	0.01 ^†^	0.03 ^†^	0.01 ^†^	0.01 ^†^	1.00	0.04 ^†^
Subacute stage group	Pre-test	13.13 ± 10.19	4.07 ± 2.41	3.07 ± 1.84	22.40 ± 10.98	0.73 ± 0.85	15.2 ± 10.02
	Post-test	16.33 ± 10.34	5.13 ± 2.83	5.27 ± 2.46	28.00 ± 10.56	0.73 ± 0.85	16.33 ± 9.43
	Mean change	3.2 ± 1.05	1.07 ± 0.77	2.20 ± 1.56	5.60 ± 3.86	0.00 ± 0.00	1.13 ± 1.75
	*p*-value	0.01 ^†^	0.01 ^†^	0.01 ^†^	0.01 ^†^	0.01 ^†^	0.03 ^†^
Chronic stage group	Pre-test	19.40 ± 2.01	7.30 ± 13.36	1.90 ± 3.59	26.30 ± 21.68	1.60 ± 1.50	13.90 ± 9.64
	Post-test	23.00 ± 22.61	9.80 ± 13.87	3.40 ± 4.27	29.40 ± 22.71	1.40 ± 1.43	15.40 ± 9.24
	Mean change	3.60 ± 3.47	2.50 ± 3.38	1.50 ± 1.11	3.10 ± 4.83	−0.20 ± 0.60	1.50 ± 2.01
	*p*-value	0.01 ^†^	0.02 ^†^	0.01 ^†^	0.09	0.34	0.05
ANOVA *p*-value		0.02 ^∞^	0.03 *	0.03 *	0.02 *	0.44	0.88

Data are presented as means ± standard deviations. Abbreviations: FMA, Fugl–Meyer Assessment; BBS, Berg Balance Scale; TIS, Trunk Impairment Scale; MBI, Modified Barthel Index; MAS, Modified Ashworth Scale; MMSE, Mini-Mental State Examination; ANOVA, Analysis of Variance; ANCOVA, Analysis of Covariance. ^†^
*p* < 0.05, paired *t*-test. * ANOVA and ^∞^ ANCOVA were performed using *p* < 0.05.

## Data Availability

The data presented in this study are available on request from the corresponding author.

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
