# Peer review of "Optimal Intervention Timing for Robotic-Assisted Gait Training in Hemiplegic Stroke"

_brainsci, 2022, doi:10.3390/brainsci12081058_

Round 1

Reviewer 1 Report

The authors should provide references for their group classification of acute, subacute and chronic stages after stroke. As known until now, the subacute phase of stroke is six months after stroke, while the chronic stage begins after six months. (line 96)https://neurolrespract.biomedcentral.com/articles/10.1186/s42466-020-00060-6 Therefore, the authors must provide scientific data regarding their group consideration or remake the statistical analysis regarding the outcomes assessed.

Result- did the authors perform the normality of data distribution before performing an independent T-test for baseline assessment? Line 197. Why did the authors use the T-test for baseline comparison and after using a non-parametric test? (lines 175-177). From line 181, I understand they assumed but not checked the distribution. The authors must be more specific and transparent regarding the statistical analysis. 

Lines 189-192. "A convenience sample of 36 patients with hemiparetic stroke (mean age: 66.86 ± 189 11.51 years; 16 females) who were admitted and received the standardized Walkbot RAGT intervention protocol between July 2018 and July 2020 at the rehabilitation hospital were retrospectively evaluated". Since it was a retrospective assessment, how was this an experimental research ( lines 99 and 147)? Please explain further or specify if this research was prospective or retrospective.

The statistical analysis must be remade by reconsidering the post-stroke phases as suggested.

Author Response

We would like to appreciate you for your precious time in reviewing our manuscript and for the information you have provided to help us better revise this paper, we have addressed all the points and provided the point-by-point responses, please see the attachment. The authors welcome further constructive comments if any.

Reviewer 2 Report

The study “Optimal intervention timing for robotic-assisted gait training in hemiplegic stroke”, a non-randomized control trial addresses a moderately original topic. The study aims to determine the best intervention time (acute, subacute, and chronic stages) for Walkbot robot-assisted gait training (RAGT) rehabilitation to improve different clinical outcomes (sensorimotor function, balance, cognition, and activities of daily living) in patients with hemiparetic stroke. The study outlines the advantages of using the RAGT in rehabilitation, contributing to the understanding of its potential in each stage.

In the first paragraph of the introduction section (lines 27-29), the authors mention that evidence suggests that treadmill-based stationary exoskeletal robotic-assisted gait training (RAGT) has benefits. However, for the acute stage, they point to only one pilot study, and for the chronic stage, they point to only one experimental study. Thus, it will be necessary to reinforce these statements with other data from the literature.

The regenerative potential after a stroke is different in each recovery phase. It would be important to include, in the introduction section, evidence that explains this phenomenon and how it might influence the measurement of the potential for using RAGT.

Figure 2 is the one mentioned first (line 149), so it would be easier to read and understand the study if the number of figures would be attributed according to the order they appear in the text.

In the flow diagram of the study (figure 2) should be stated the specific reasons for de exclusion of 11 potential participants. Additionally, in the paragraph (lines 192-194)) Because of the complicated nature of patients with stroke, only 36 (76%) of the 47 patients who successfully completed the pre-test, intervention, and post-test were included in the final data analysis. The dropout reasons included sudden discharge, medical complications, and fatigue.” Is stated that some of the eleven patients did start the assessment and intervention but dropout during the study. Thereby, these participants should not be considered exclusions at the beginning but dropouts during the intervention. The authors should clarify this information in the text and in the diagram.

For all the assessment instruments (lines 105-145) should be described or referred the original/ development source of evidence and the most recent evaluation of validity and reliability.

The authors could explain the decision to start the procedures with a 0.5 km/h gait velocity (line 165)?

Were the verbal encouragements standardized for all patients (line 166)?

As it was used to assess the study results four instruments other than BBS and MBI, the authors didn’t consider the need to calculate the sample size for this specific study instead of using an auto-referenced previous one?

As the study's authors mentioned that FMA outcome measurement was statistically different among the groups.” (line 184-185), in the tables 1 or 2, it should be reported the results between groups in the pre-test assessment.

The symbols used in table 2 (p < 0.05, paired t-test or Wilcoxon signed-rank test. * ANOVA or ANCOVA was performed using p < 0.05.) should be different for each test, and all the exact p values should be presented.

Line 212 "This suggests greater improvements in sensorimotor recovery after the Walkbot RAGT intervention in the ASG." The interpretation of the results should be performed at Discussion section.

The quality of figure 3 should be improved.

The affirmation “Combined with this study, it can be concluded that RAGT is efficient in improving motor function recovery in patients with hemiparetic stroke, particularly in the acute phase.” (line 286-288) Should be reformulated as the study is not a randomized control trial and has several limitations that prevent the assumption of definite conclusions.

In the discussion section, although authors addresses that “research primarily focuses on the effectiveness of Walkbot RAGT in improving patients’ motor function compared with the effectiveness of CPT” (lines 348-349) authors didn’t discuss on whether the recovery rate and the difference between the groups would be similar if only conventional physical therapy were applied. It should be included a brief reflection on the topic.

Also in the discussion section it should be addressed the fact that were included in all groups both Ischemic and Hemorrhagic types of stroke and if the results should be interpreted accordingly.

Author Response

(The authors gave the same response as above.)

Round 2

Reviewer 1 Report

The authors accomplished the suggestions I made for the paper. I consider is suitable for publication.

This manuscript is a resubmission of an earlier submission. The following is a list of the peer review reports and author responses from that submission.

Round 1

Reviewer 1 Report

This study is an open trial where motor and cognitive outcomes are reported after RAGT at different stages of stroke recovery (acute, subacute, chronic).

The main concern is that the superiority of recovery during the acute stage is easily related to the spontaneous recovery that occurs in the first weeks after stroke and would not depend on the type of training (RAGT). The Cochrane review has previously established the conclusion that patients with stroke in their subacute stage of recovery would benefit more from RAGT (Mehrholtz et al. 2020 https://pubmed.ncbi.nlm.nih.gov/33091160/).

Moreover, the study presents several general methodological flaws:

- the research protocol was retrospectively registered (this is not stated in the section)

- the selected outcome measures are not fully exhaustive (a measure of gait dependency/disability such as FAC has not been used)

- the sample size is too small to drive any conclusion or to generate a hypothesis

 Other majors' specific issues:

Introduction:

1.       This section should be extensively reviewed, including more recent papers (i.e.) on RAGT and their role in stroke recovery (in particular acute-subacute stage)

2.       The hypothesis on the effects of RAGT on cognition has not been fully explained

Methods

1.       Study design: prospective or retrospective? The protocol was registered on 2021

2.       It’s not clear the disability level of the patients enrolled (all dependent gait?)

3.       Fig.1: should be placed in the results section; moreover, several data are missing: any patients excluded from this study? In the protocol is reported that 47 patients were recruited. Please clarify. Any drop-outs? (https://trialsearch.who.int/Trial2.aspx?TrialID=KCT0006333)

 Results

1.       No information on the severity of gait impairment is reported in Table 1. Are they similar at baseline for gait impairment?

2.       Looking at Table 2 seems that acute strokes were less impaired (FMA 30) compared to subacute (13) or chronic (19). The severity of gait impairment can be another independent factor influencing recovery and responsiveness to RAGT.

Discussion

This section is difficult to follow in several parts and should be extensively revised. A digression on neuroplasticity has been done without a clear relation with the reported results. Moreover, the conclusions are not supported by the results and the study design.

The references list is not updated and entirely relevant (i.e. the Cochrane review has not been mentioned)

Reviewer 2 Report

In this study Lingchao Xie and colleagues investigated in which phase of the stroke the use of an exoskeleton for gait rehabilitation can be more effective. The aim of the study is interesting and the study is appropriately structured. However, there is a concern regarding the different characteristics of recovery after a stroke (especially in mixed samples with hemorrhagic and ischemic stroke). Usually, in the acute and subacute phase, an exponential trend in the functional recovery is expected. Differently in the chronic phase the trend is different and the functional gain is limited. “Large-scale observational studies indicate that recovery is most rapid during the first month after stroke, and motor function typically reaches a plateau within 3 months” (Stinear et al., 2014). This aspect can affect the results and the conclusion when the difference in the pre-post treatment scores are compared (in particular) between acute and chronic phases. In my opinion, this is an important key-point to be discussed in the limitations paragraph. In the conclusion paragraph can be useful to explain that the result of the study suggests starting therapy early.

Line 15 (a): The FMA is composed of two parts. One for the upper extremity and one for the lower extremity. In the text it is not clear if the authors use the complete version of the tool. Observing the mean score of results, I think that the authors used the FMA only for the lower limb. Please clarify in the text, you can use the most common (FMA-LE). Differently, the results of this session of the scale are useful in a trial that investigate the gait function; please provide them.

Line 15 (b): Although the choice of a multi-domains evaluation approach is a strength of the study, the order of the battery tools should first focus on the aspects directly involved in the selected rehabilitation protocol. The cognitive function can be indirectly affected from the mobility but is not a primary outcome of investigation. I think it is appropriate to begin the list (and consequently the order in all the rest of the paragraphs) by reporting (for example) first the two outcomes selected for power analysis, followed by the motor ones and finally the cognitive one.

Line 18: “The analysis of variance showed significant group main effects on FMA, BBS, MBI, and TIS variables between the ASG and the SSG or CSG (p < 0.05).”. This sentence is not totally clear, please re-write.

Line 30: “Currently, the Walkbot ankle-knee-hip-controlled RAGT model (P&S Mechanics, Seoul, Republic of Korea) and Lokomat knee-hip-controlled RAGT model (Hocoma, Volketswil, Switzerland) have been commonly utilized, and their clinical efficacy has been well investigated in the acute, subacute, or chronic stages of recovery [4-6]”. The study performed the investigation using the “Walkbot”. In this sentence it is better to name only the device used in the current investigation or to change with the “exoskeleton devices” in general and report the names inside brackets. 

Line 37: “gait” is retoundance.

Line 88: The authors reported “International Clinical Trials Registry Platform (KCT0006333)”. I’ve not found the registration on the dataset of the International CTRP Please provide more details or a direct link to find the registration.

Line 135: “The three groups received 30 min of therapy, three times a week, for four weeks (12 sessions in total, with a minimum of eight sessions)”. Observing the week frequency, four treatments missing indicate that the subject has undergone more than one week less treatment. Please provide the mean and standard deviation of sessions for each group and explain why 8 sessions was chosen like a cut-off.

Line 319: “Furthermore, recovery was correlated with increased dendritic branching of layer V motor cortex [...]” use “the” before “recovery”.

Line 359: Please provide the date of ethical approval
